# A protocol for using human genetic data to identify circulating protein level changes that are the causal consequence of cancer processes

Lisa M. Hobson[1,2]*, Richard M. Martin[2,3], Karl Smith-Byrne[3,4], George Davey Smith[1,2,3], Gibran Hemani[1,3], Joseph H. Gilbody[1,2], James Yarmolinsky[1,2,5], Sarah E.R. Bailey[6], Lucy J. Goudswaard[1,2]*, Philip C. Haycock[1]*

1 MRC Integrative Epidemiology Unit, Bristol Medical School, University of Bristol, Bristol, United Kingdom, 2 Population Health Sciences, Bristol Medical School, University of Bristol, Bristol, United Kingdom, 3 Bristol NIHR Biomedical Research Centre, University Hospitals Bristol and Weston NHS Foundation Trust and the University of Bristol, Bristol, United Kingdom, 4 Cancer Epidemiology Unit, Nuffield Department of Population Health, University of Oxford, Oxford, United Kingdom, 5 Department of Epidemiology and Biostatistics, School of Public Health, Imperial College London, London, United Kingdom, 6 Department of Health and Community Science, University of Exeter, Exeter, United Kingdom

* Lisa.hobson@bristol.ac.uk (LMH); lucy.goudswaard@bristol.ac.uk (LJG); philip.haycock@bristol.ac.uk (PCH)

## Abstract

### Introduction

Cancer is a leading cause of death worldwide. Early detection of cancer improves treatment options and patient survival but detecting cancer at the earliest stage presents challenges. Identification of circulating protein biomarkers for cancer risk stratification and early detection is an attractive avenue for potentially minimally invasive screening and early detection methods. This research aims to identify protein level changes that are downstream of genetic liability to lung cancer and colorectal cancer.

### Methods and analysis

PRS will be calculated using the PRS continuous shrinkage approach (PRS-CS and PRS-CSx) for colorectal and lung cancer risk. This methodology utilises effect sizes from summary statistics from genome-wide association studies (GWAS) available for the cancers of interest to generate weights via the continuous shrinkage approach which incorporates the strengths of the GWAS associations into the shrinkage applied. This methodology both improves upon previous PRS methods in accuracy as well as improving cross-ancestry application in the PRS-CSx approach. GWAS summary statistics will be from the Genetics and Epidemiology of Colorectal Cancer Consortium (GECCO) and the International Lung Cancer Consortium (ILCCO). The association between the polygenic risk scores and 2923 proteins measured by the Olink platform in UK Biobank (UKB) participants with protein measures available will

**Data availability statement:** No datasets were generated or analysed during the current study. All relevant data from this study will be made available upon study completion.

**Funding:** LMH is supported in part by grant MR/N0137941/1 for the GW4 BIOMED MRC DTP, awarded to the Universities of Bath, Bristol, Cardiff and Exeter from the Medical Research Council (MRC)/UKRI RMM is a National Institute for Health Research Senior Investigator (NIHR202411). RMM, LJG and PCH are supported by a Cancer Research UK 25 (C18281/A29019) programme grant (the Integrative Cancer Epidemiology Programme). RMM is also supported by the NIHR Bristol Biomedical Research Centre which is funded by the NIHR (BRC-1215-20011) and is a partnership between University Hospitals Bristol and Weston NHS Foundation Trust and the University of Bristol. Department of Health and Social Care disclaimer: The views expressed are those of the author(s) and not necessarily those of the NHS, the NIHR or the Department of Health and Social Care. SERB was supported by an NIHR Advanced Fellowship (NIHR 301666) whilst undertaking this work. Additional support was provided by the Higgins family. The funders had no role in study design, data collection and analysis, decision to publish, or preparation of the manuscript.

**Competing interests:** RMM, LJG and PCH have received funding from Cancer Research UK, a commercial funder. LMH receives funding from the GW4 BioMed2 MRC DTP, a commercial funder. This does not alter our adherence to PLOS ONE policies on sharing data and materials.

be assessed using linear regression under the assumption of linearity in the proteomic data. The proteins identified could represent several different scenarios of association such as forward causation (protein causes cancer), reverse causation (cancer genetic liability causes protein level change), or horizontal pleiotropy bias (no causal relationship exists between the protein and cancer). Forward and reverse Mendelian randomization sensitivity analyses, as well as colocalization analysis, will be performed in efforts to distinguish between these three scenarios. Protein changes identified as causally downstream of genetic liability to cancer could reflect processes occurring prior to, or after, cancer onset. Due to individuals in the UKB having proteins measures at only one timepoint, and because UKB contains a mix of incident and prevalent cases, some protein measures will have been made prior to a cancer diagnosis while others will have been made after a cancer diagnosis. We will explore the strength of association in relation to the time between protein measurement and prevalent or incident cancer diagnosis.

## Introduction

Detecting cancer at an early stage is important because patients diagnosed early have a greater chance of being treated with curative intent and so experience increased long-term survival. Cancer is a leading cause of death worldwide [1] with survival rates considerably lower when diagnosis is made at a later stage. For colorectal cancer and lung cancer the 5-year survival is reduced from more than 9 in 10 and 6 in 10, respectively, when diagnosed at stage 1, to 1 in 10 for colorectal cancer and less than 1 in 10 for lung cancer when diagnosed at stage 4 [2]. The NHS Long Term Plan aims to increase early detection of cancers from half to three quarters by the year 2028 to improve cancer survival [3] as currently in England only 54% of cancers are detected early [4]. To meet this goal a number of research challenges need to be addressed, including development of methods for determining cancer risk (i.e., risk stratification) and identifying biomarkers which are effective for detecting cancer at an early stage [5].

One of the challenges to be overcome in improving cancer early detection is the identification of specific biomarkers for the cancers of interest that can be measured by minimally invasive, low-cost methods and are able to be implemented in a clinical setting. One way to address this challenge is the measurement of circulating protein levels in blood serum or plasma, potentially feasible because of the widespread use of blood tests in healthcare. Circulating protein biomarkers are a potentially useful tool for several clinical areas including identifying groups at high risk of the future development of cancer (risk stratification), early detection, disease diagnosis and monitoring biological processes [6,7]. They may provide a minimally invasive means of screening asymptomatic individuals for undiagnosed disease or for diagnosis of symptomatic patients [8,9]. An advantage of measuring protein within the blood is the reduced volume of sample required for analysis and affordability, as opposed to other minimally invasive methods, such as circulating-tumour DNA (ctDNA) sequencing.

For the Olink explore 3072 panel, only 6 μL of plasma or serum is required vs. 4-5mL of plasma to obtain 5–10ng/mL of ctDNA and with the possibility of implementation of protein testing via ELISA, costing around ~£4 per test [10–15].

One approach to biomarker discovery is via prospective cohort studies to identify proteins associated with the incidence of a disease of interest, by measuring protein levels in individuals before diagnosis. These methods require large sample sizes over long periods of time to capture these events, at great financial and time cost [16]. A comparatively inexpensive technique for biomarker discovery has been formalised by Holmes and Davey Smith [17], and involves application of Mendelian randomization (MR) of disease liability as the exposure on protein levels as the outcome (sometimes described as reverse MR or reverse gear MR). Building on this idea, we propose that protein level changes resulting from cancer onset can be identified via an individual's PRS for specific cancers, representing their genetic liability to developing that cancer. Defining the point of "cancer onset" remains difficult, with many possible mechanisms of initiation; for the purpose of this study we will use date of diagnosis to determine prevalent vs. incident cases within the cohort [18].

Proteins associated with genetic liability to cancer could reflect different mechanisms of association. Associations could reflect 'forward causation' where the protein is upstream of and causal for cancer, e.g., P1-6 (Fig 1, panel A) or 'reverse causation' where carcinogenesis is causing the change in downstream protein level, e.g., P7-9 (Fig 1, panel A). Proteins that are associated via forward causation are upstream of the cancer pathway and therefore do not always denote the presence of cancer but could identify potential therapeutic targets for cancer prevention and cancer prediction, these protein levels will likely remain stable over long periods of time. Proteins downstream of cancer development will likely show more variation in levels resulting from the progression of cancer; we will refer to these proteins as "reverse causal". For proteins that cause cancer, most of the variants in the cancer PRS will have no causal relationship to those proteins, e.g., G9 and P6 (Fig 1, panel B), whereas for proteins that are causally associated downstream of cancer liability pathways, all variants in the PRS could contribute to the association signal (Fig 1, panel C). We thus expect a cancer PRS to be better powered for discovery of proteins downstream of cancer development. However, the relative balance in findings reflecting scenarios one ('forward causation') and two ('reverse causation') is likely to depend on the prevalence of cancer, including early or pre-clinical stages, in the sample used to measure the proteins. In general, the higher the prevalence, the greater the number of associations we expect to see reflecting effects of cancer liability pathways on protein concentration. Association of proteins with a genetic liability to cancer can also be due to factors other than genetic liability such as horizontal pleiotropy bias, as illustrated in Fig 1, panel A by G4 and G6; which may negate its use in risk stratification or early detection.

## Aims and objective

Cancer early detection remains a challenge, with a lack of specific biomarkers for lung and colorectal cancer. This research aims to identify protein level changes that are causally downstream of genetic liability to lung cancer and colorectal cancer for use as potential biomarkers for these cancers. To achieve this, we will use a combination of PRS, observational analyses, bidirectional Mendelian Randomization and colocalization approaches.

## Methods and analysis

### Polygenic risk score analyses

PRS can be developed using GWAS summary statistics on the associations of many SNPs across the genome with cancer. In this way millions of SNPs can be combined to develop an individual's PRS for a cancer. A PRS is the sum of the number of copies of risk alleles individuals have for SNPs across the genome, weighted by the effect size of these SNPs in relation to the disease of interest, in this case, cancer [19]. While the initiation of cancer and the factors that contribute to the onset and progression of cancer are still not fully understood [20]; by calculating a PRS using data from all SNPs across the genome, SNPs involved in initiation, promotion and progression of cancer will be captured by this score, reflecting the complex process of cancer development [18].

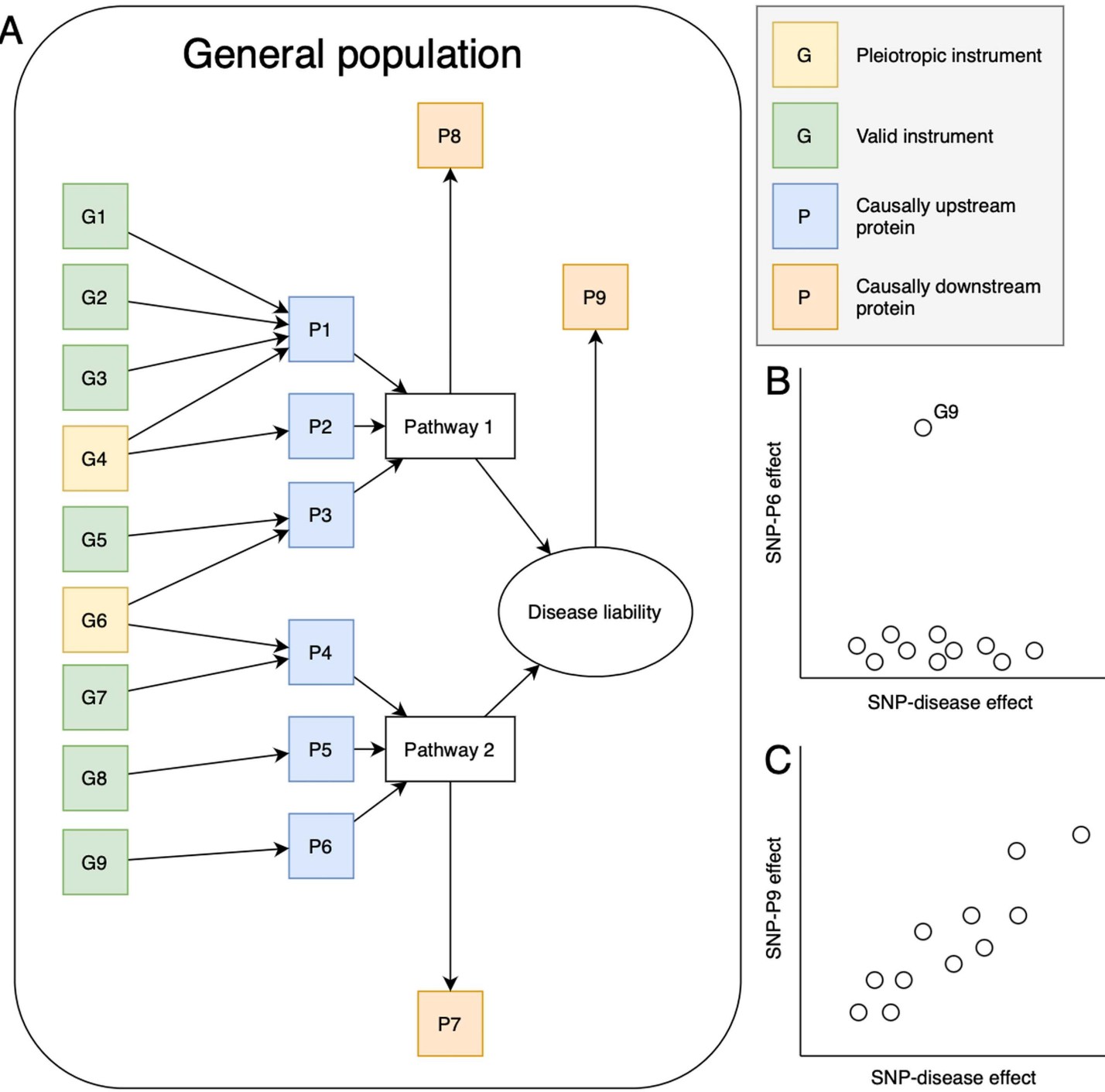

**Fig 1. Illustration of the difference pathways which may be reflected by protein associations with genetic variables and disease liability.** Panel A shows the relationship between genetic variants (G1-9), upstream proteins (P1-6) and downstream proteins (P7-9) via the disease liability pathways. Panel B shows the relationship between SNPs and upstream protein P6. Panel C shows the relationship between SNPs and downstream protein P9.

In this study, PRS will be calculated for UKB participants (Application ID: 15825/81499) with proteomic measurements (N = 49,542), individuals with sex-mismatch (derived by comparing genetic sex and reported sex) or individuals with sex-chromosome aneuploidy will be excluded from the analysis (N = 814) as well as highly related individuals related to a 3rd degree to >200 individuals (N = 2) [21]. For colorectal cancer, we will use effect weights derived from GWAS summary statistics of the: i) Genetics and Epidemiology of Colorectal Cancer Consortium (GECCO) (GWAS Catalog Accession: GCST90255675) for Europeans; and ii) the Asia Colorectal Cancer Consortium (ACCC)/Korean-National Cancer Centre CRC Study 2 (Korea-NCC2) for GWAS summary statistics for East Asians. For lung cancer, we will use effect weights derived from GWAS summary statistics from the International Lung Cancer Consortium (ILCCO) (GWAS Catalog Accession: GCST004748). Sample selection and quality control within these studies has been previously described [22,23].

PRS will be derived from the PRS-CS approach, using summary statistics from GWAS for the cancer of interest along with an external linkage disequilibrium (LD) reference panel corresponding to the ancestry of the GWAS. The continuous shrinkage approach incorporates the strengths of the GWAS associations into the shrinkage applied to shrink small SNP effects towards zero, while large effects are unaffected [24], generating a posterior effect size for each SNP [25]. These weights will be used to calculate the PRS of UK Biobank participants for colorectal and lung cancer, calculating the sum of risk increasing alleles across all genetic variants weighted by the effect sizes generated by PRS-CS [25,26]. PRS-CSx applies the same methodology as PRS-CS to multi-ancestry GWAS summary statistics, improving generalisability of results to more ancestry groups within the global majority by allowing the use of different ancestry GWAS summary statistics and LD panels for those populations [27]. In an effort to reduce the Eurocentric bias and to increase power in addition to developing a European PRS, we will be utilising GWAS summary statistics for colorectal cancer from European and East Asian ancestries to develop polygenic risk scores [22,28,29].

## Cancer subgroup analyses

In addition to a PRS for overall colorectal and lung cancers, we will calculate a PRS for colon cancer and rectal cancer specifically and for lung cancer subgroups (adenocarcinoma, squamous cell carcinoma, small cell carcinoma). Additionally, PRS scores will be calculated for never smokers and ever smokers using weights generated from summary statistics of GWAS for lung cancer in never smokers and lung cancer in ever smokers.

## Olink proteins

Olink protein measurements were performed as part of the Pharma Proteomic Project (UKB-PPP) on blood plasma samples using the antibody-based protein Olink Explore 3072 Proximity Extension Assay. Proteomics were generated for 54,219 participants who were considered to be highly representative of the UK Biobank population on baseline characteristics and showed enrichment for selected diseases [30]. The number of participants with colorectal and lung cancers can be seen in Table 1. Quality control, sample selection and data processing has been described previously [30]. Associations between the participants' PRS and 2923 Olink protein measures from the UK Biobank will be tested via linear regression, adjusting for age, sex, principal components and sample storage time where this has an impact on protein level variation [31]. Protein measures will undergo inverse rank normal transformation (INT) for each protein [32]. The number of independent proteins will be calculated using the metaboprep R package [33]. False discovery rate correction will be applied to p-values, proteins with p-value less than the calculated alpha will be prioritised for further analyses.

## Sensitivity analyses

Proteins identified from association analyses may reflect different scenarios, including causation or confounding from population stratification or dynastic effects. Some possible scenarios include: (1) a protein may be a cause of cancer risk, which we define as "forward causation"; (2) an alternative scenario is that the protein identified is causally downstream

**Table 1. Cancer frequency within the UKB cohort and within the UKB-PPP study participants.**

| ICD10/ICD9 code | Cancer | Number of cases (UKB) | Number of cases (UKB-PPP) |
|---|---|---|---|
| C18/153 | Malignant neoplasm of colon | 6015 (1.2%) | 630 (0.13%) |
| C19/1540 | Malignant neoplasm of rectosigmoid junction | 642 (0.13%) | 67 (0.013%) |
| C20/1541 | Malignant neoplasm of rectum | 2670 (0.53%) | 268 (0.053%) |
| C34/162 | Malignant neoplasm of bronchus and lung | 5305 (1.1%) | 597 (0.12%) |
| **ICD-O-3 Code** | **Histological Subset** | **Number of cases (UKB)** | **Number of cases (UKB-PPP)** |
| 8140, 8211, 8250–8260, 8310, 8323, 8480–8490, 8550 | Lung adenocarcinoma | 2390 (0.48%) | 261 (0.052%) |
| 8070-8072 | Lung squamous cell carcinoma | 1244 (0.25%) | 151 (0.03%) |
| 8041–8042 | Lung small cell carcinoma | 502 (0.1%) | 64 (0.013%) |

Number of cases for each cancer type derived from UK Biobank phenotypic data with percentage of cases out of overall individuals is represented in brackets. UKB overall n = 501939, UKB with protein measures (UKB-PPP) n = 52996.

of cancer liability, which we refer to as "reverse causation"; (3) there is no causal relationship between the protein and cancer and the identified association reflects horizontal pleiotropy, (4) due to population stratification where spurious associations are due to differences in the GWAS population and those that the PRS is calculated on. We will perform various sensitivity analyses to distinguish amongst these scenarios, described below.

### Bidirectional Mendelian randomisation sensitivity analyses

MR uses genetic variants, associated with the phenotype of interest as the instrumental variable to assess the effect of the phenotype on an outcome. Due to the random nature of inheritance of genetic variants there is an advantage over observational epidemiology whereby confounders may influence both the exposure and outcome of interest [34–36]. Genetic associations used in MR analyses often come from GWAS summary data, whereby association is conventionally defined by a p-value threshold of $5 \times 10^{-8}$.

   **Assumptions.** The three core assumptions of MR, known as the instrumental variable (IV) assumptions (Fig 2), are relevance (IV1) – is the instrumental variable (G) associated with the exposure (E), independence (IV2) – there is no confounding of the association between the instrument (G) and outcome (O) (this can arise through population stratification, dynastic effects and assortative mating) and exclusion restriction (IV3) – the instrumental variable (G) does not act on the outcome (O) except via the exposure (E) no horizontal pleiotropy (red dashed line, IV3) [37–39].

### Study design

MR will be performed in the forward and reverse direction: forward MR, where the protein is the exposure and cancer is the outcome, will be used to estimate the effect of selected proteins on the cancer of interest and reverse MR, where cancer is the exposure and protein levels are the outcomes, will be used to estimate the effect of cancer liability on circulating protein concentration [17]. Forward MR will be performed using cis-pQTLs to instrument proteins identified as being associated with the cancer PRS, the threshold for these will be $p < 3.4 \times 10^{-11}$ [40]. Cis-pQTLs will be defined as within < 1Mbp of the protein coding gene and trans-pQTLs will be defined as > 1Mbp away from the protein coding gene [41]. Reverse MR will be performed using SNPs associated with the cancer PRS at a threshold of $p < 5 \times 10^{-8}$ excluding cis-pQTLs for the protein. If association is found in the forward direction this may suggest that the protein is causal for the cancer but if association is found in the reverse direction this may suggest that genetic liability to cancer is causing the protein level change [17]; to elucidate this causality, different MR estimation methods will be employed, the application and conditions of these are described below.

IV1

IV2

IV3

**G** – *Instrumental variable (proxy)*
**E** – *Exposure*
**O** – *Outcome*
**C** – *Confounder*

**Fig 2. Directed Acyclic Graphs (DAGs) showing the three IV assumptions of Mendelian Randomization.** IV1 represents that the assumption that the genetic variant (G) is strongly associated (red arrow) with the exposure (E). IV2 represents the assumption that confounders don't also act on the genetic variant (red dashed arrow) as they do the outcome. IV3 represents the assumption that the genetic variants do not affect the outcome through other routes separate from the exposure (red dashed arrow) (C).

## Instrument & method selection

The strength of instrument will be determined by calculating the F-statistic, a measure of potential weak instrument bias that could arise from the use of IVs as a proxy for the effect of exposure on outcome [39]. The F-statistic takes into account the genetic variance ($R^2$), the sample size and how many instruments are present. An F-statistic greater than 10 indicates that the bias from weak instruments is small, where this F-stat is less than 10 this indicates a possibility of bias and will be noted [42].

Dependent on the number of SNPs available, the appropriate method of effect-estimation will be selected for MR analyses. For proteins with a single pQTL SNP the Wald ratio will be calculated as the ratio of SNP-outcome/SNP-exposure association [43]. For proteins with two or three independent SNPs, a fixed-effects inverse variance weighted (IVW) model will be used. For four or more independent SNPs, a random effects inverse variance (IVW) model, combining multiple SNP outcome/exposure Wald ratio, will be used [44]. In the event of multiple independent SNPs, pleiotropy will be considered by calculating Cochran's Q statistic, a method for assessing global and individual pleiotropy across instruments [45]. Weighted mode and weighted median methods will also be used when > 10 SNPs are available [46,47].

The MR-PRESSO, weighted mode and weighted median methods will be used to assess IVs for horizontal pleiotropy, violation of IV3 where the IV acts on the outcome not via the exposure, by comparing estimates with and without suspected pleiotropic variants, this will be repeated for both forward and reverse MR [48]. In addition being robust to

pleiotropy methods such as MR robust adjusted profile score (RAPS) also accounts for other potential sources of bias such as weak instruments and measurement error in the exposure [49]. MR-CAUSE (Causal Analysis Using Summary Effect estimates) is another method that can be used when IV3 is violated due to pleiotropic effects of correlated pleiotropy, where the pleiotropic factor is a confounder of the exposure-outcome association versus when the IV has effects on pleiotropic factor independently of the effect of the IV on the exposure this is uncorrelated pleiotropy [50].

When performing reverse MR using a larger numbers of SNPs, clustered heterogeneity can occur when different genetic variants are causally associated via distinct pathways. To assess this, clustering based methods can be used to divide groups based on these estimates of causality [51]. MR-Clust will be used to investigate clustered heterogeneity across IVs and identify potential distinct pathways that make up the effect estimate; clustering works by separating the variants into clusters with additional null and junk clusters, representing no causal effect or those that do not fit within the distinct clusters [51,52]. Another clustering method that will be used is the Noise-Augmented von Mises-Fisher Mixture model (NAvMix), this method allows for variants to belong to multiple clusters based on their probability of membership to that cluster [53,54]. The contamination mixture model method can also be used to cluster into distinct groups based on the IVs causal effect estimate even when invalid IVs are present [55]. PheWAS-based clustering will also be used to cluster SNP associations based on different pathways and thus help identify other causal pathways of the PRS – protein associations found [56]. The methods of MR described make different assumptions and aim to address different violation of the IV assumptions, testing of these different methods has illustrated the variation in accuracy and the need for appropriate method selection based on the datasets used [57].

### Data harmonisation

Harmonisation across the GWAS summary statistics datasets will be performed using the TwoSampleMR R package to ensure that the effect on exposure and outcome are in the same direction, using effect allele frequency (EAF) to infer the strand, palindromic SNPs are kept where they are inferable from EAF. [58].

### Colocalization

Associations may be due to genomic confounding, where genetic variants in linkage disequilibrium (LD) at the same locus act on the cancer and protein via separate pathways, a form of horizontal pleiotropy bias. Colocalization analyses will be used to assess if genetic associations with cancer and proteins are due to shared causal variants at the same locus through genomic confounding [59,60].

A full overview of the sample selection and filtering of available data is shown in Fig 3, including the number of participants after filtering and results to be taken forward.

### Ethics

The colorectal cancer GWAS conducted by Fernandez-Rozadilla et al. (2022) was approved by the South Central Ethics Committee (UK) under the reference number 17/SC/0079 [22].

All studies used in the lung cancer GWAS conducted by McKay et al. (2017) obtained local ethics committee approval and all participants gave informed consent [23].

Application for colorectal cancer site specific GWAS summary statistics from GECCO has been approved.

Application for summary statistics from the Asian Colorectal Cancer Consortium (ACCC) and the Korean-National Cancer Center CRC Study 2 (Korea-NCC2) will be submitted.

UK Biobank was approved by the North West Multi-centre Research Ethics Committee (MREC) as a Research Tissue Bank (RTB) approval renewed in 2021, all participants in the study have given informed consent [61]. Genotype, phenotype and Olink protein measure data access has been obtained under Application ID: 15825/81499.

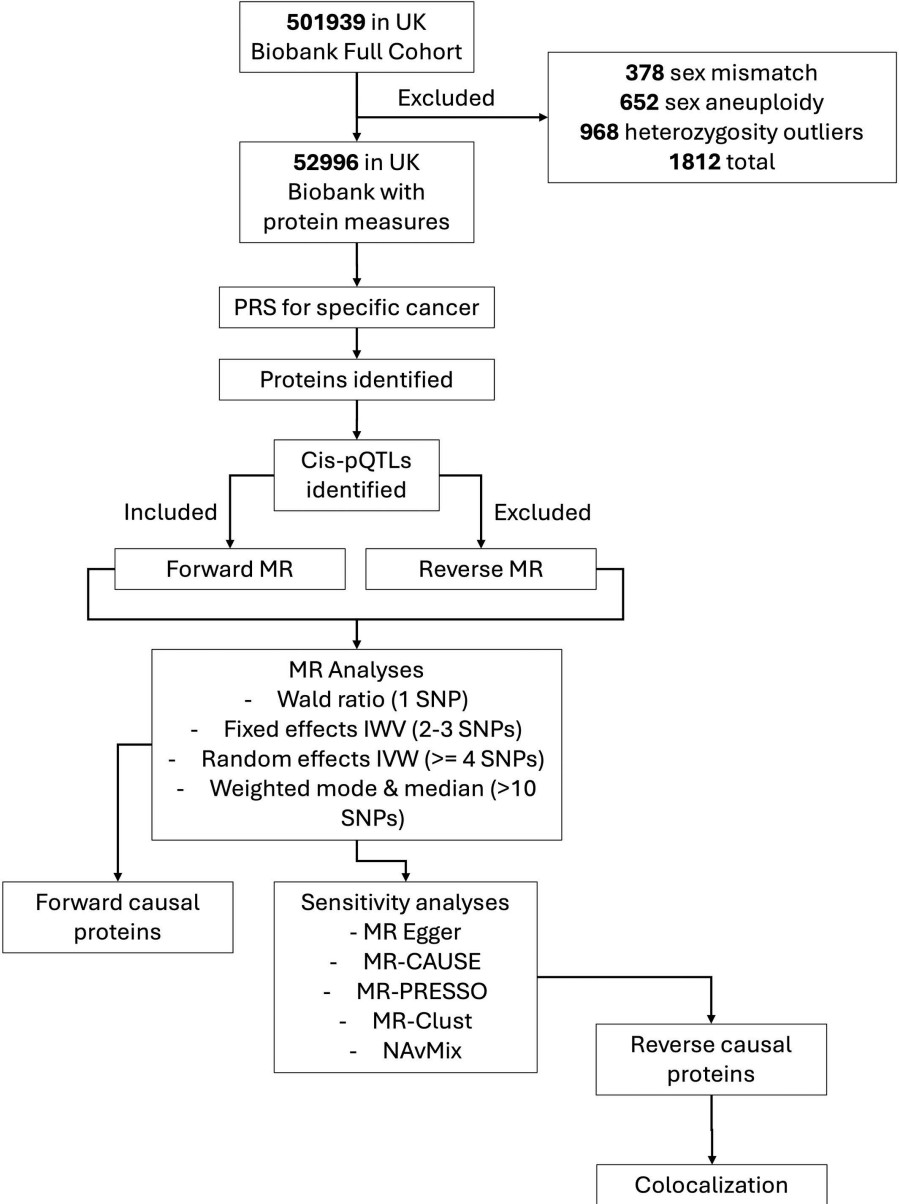

**Fig 3. Flowchart of methodology and data selection.** UK Biobank data will be filtered based on exclusion criteria and availability of protein data, this will filter into proteins identified. Cis-pQTLs for the proteins will be identified which will either by used as instruments for forward MR or excluded from reverse MR. Various MR analyses will be performed based on number of SNPs available and sensitivity analyses will be used to filter proteins to perform colocalization.

## Further analyses

**"Time-to" and "Time-from" diagnosis.** In observational analyses, we will evaluate the magnitude of the relationship between proteins, taken either pre or post-diagnosis, and cancer risk. This will involve an analysis of prevalent and incident cancer cases in UKB (Table 2) and a time variable derived from date of cancer diagnosis and time of blood collection [62,63]. To adjust for any variation in protein concentration as a result of sample storage and protein

**Table 2. Prevalent and incident cases of cancer from UKB cohort and within the UKB-PPP study participants.**

| ICD10/ICD9 Code | Cancer | UKB Overall | | UKB-PPP | |
|---|---|---|---|---|---|
| | | Prevalent Cases | Incident Cases | Prevalent Cases | Incident Cases |
| C18/153 | Malignant neoplasm of colon | 1029 | 4986 | 97 | 533 |
| C19/1540 | Malignant neoplasm of rectosigmoid junction | 186 | 456 | 22 | 45 |
| C20/1541 | Malignant neoplasm of rectum | 574 | 2096 | 60 | 208 |
| C34/162 | Malignant neoplasm of bronchus and lung | 245 | 5060 | 31 | 566 |
| ICD-O-3 Code | Histological Subset | Prevalent Cases | Incident Cases | Prevalent Cases | Incident Cases |
| 8140, 8211, 8250–8260, 8310, 8323, 8480–8490, 8550 | Lung adenocarcinoma | 77 | 2313 | 8 | 253 |
| 8070-8072 | Lung squamous cell carcinoma | 74 | 1170 | 14 | 137 |
| 8041–8042 | Lung small cell carcinoma | 23 | 479 | 3 | 61 |

Number of incident and prevalent cases for each cancer type derived from UK biobank phenotypic data.

degradation over time [31], the relationship between storage time and protein level for all protein measures available will be assessed. Proteins are more likely to be causally downstream of cancer onset if the association with cancer is sensitive to time between protein measure and cancer diagnosis, a potential route for differentiating between normal baseline levels and levels that suggest the presence of cancer. If protein levels are detectable prior to patient reported symptoms proteins may be more suited for screening and early detection.

**Replication of findings.** Replication of protein association and MR will be carried out in the DECODE cohort [64] and EPIC study [65] where proteins are available.

## Software

This work will be carried out using the computational facilities of the Advanced Computing Research Centre, University of Bristol - http://www.bristol.ac.uk/acrc/.

PRS-CS (https://github.com/getian107/PRScs) and PRS-CSx (https://github.com/getian107/PRScsx) will be used to calculate polygenic risk scores, using R, Python and PLINK.

Metaboprep (https://github.com/MRCIEU/metaboprep) R package will be used to calculate independent proteins [33]. Mendelian Randomization analyses and data harmonisation will be performed using the R packages TwoSampleMR (https://github.com/MRCIEU/TwoSampleMR) and MendelianRandomization (https://cran.r-project.org/web/packages/MendelianRandomization/index.html) [66].

Proteins will be inverse rank normal transformed using the "RankNorm" function in R package "RNOmni" (https://cran.r-project.org/web/packages/RNOmni/index.html) [32].

## Patient and public involvement

A summary of the proposed research was presented to members of a patient and public involvement group, with either personal experience with cancer or experience via a family member. The feedback received was that this was very important research and that they believe it would be useful for early detection for cancers that do not yet have specific screening via a blood test. Updates about this study will also be disseminated to the group.

Results of these analyses will be disseminated via the University of Bristol MRC Integrative Epidemiology Unit IEU Portal and submitted as a manuscript to a peer-reviewed journal for publication. All statistical code will be made available via GitHub.

Polygenic risk scores and PRS-protein associations will be returned to the UK Biobank in line with the UK Biobank obligation for researchers outlined [67].

**Strengths and limitations of this study**

- Strengths of the study:

  ◦ A strength of using PRS in the discovery step of identifying proteins to carry through into MR, is that lifetime genetic liability to cancer is captured with the use of a continuous measure of genetic risk. In contrast, using cancer registry data only captures those who already have a diagnosis and not those who may go on to be diagnosed imminently.

  ◦ This study will use a novel approach to construct PRS using weights generated from SNPs across the genome, we expect this to be more powerful for discovery in comparison with a PRS constructed using only genome-wide significant SNPs.

- Limitations of the study

  ◦ Lack of protein data for diverse population groups within available datasets; therefore, results may not be generalisable to ancestries outside of the European population for whom sufficient protein data was available for this study.

  ◦ UKB participants reflect a subset of the population that volunteered for the study, participants are from a higher socioeconomic position than average which could introduce collider bias.

  ◦ Prevalent cancer cases will reflect a specific subset of the general population with cancer, individuals who have survived cancer and were able to volunteer for the study; potentially introducing survivorship bias.

  ◦ It cannot be ruled out that proteins may reflect effects of processes beyond cancer liability to protein pathways.

  ◦ Lack of staging information for cancer cases within the UKB limiting our ability to distinguish early versus more advanced cancers.

  ◦ The proteomic technology currently used measures protein binding as opposed to protein levels

  ◦ Subgroup analyses are more prone to false positive results which will require replication analysis to ensure these reflect actual associations.

  ◦ Power calculations have not been performed to assess the power of the subgroup analyses.

## Author contributions

**Conceptualization:** Richard M. Martin, Karl Smith-Byrne, Sarah E.R. Bailey, Lucy J. Goudswaard, Philip C. Haycock.

**Methodology:** Richard M. Martin, Karl Smith-Byrne, Sarah E.R. Bailey, Lucy J. Goudswaard, Philip C. Haycock.

**Supervision:** Richard M. Martin, Karl Smith-Byrne, Sarah E.R. Bailey, Lucy J. Goudswaard, Philip C. Haycock.

**Visualization:** Gibran Hemani.

**Writing – original draft:** Lisa M. Hobson.

**Writing – review & editing:** Richard M. Martin, Karl Smith-Byrne, George Davey Smith, Gibran Hemani, Joseph H Gilbody, James Yarmolinsky, Sarah E.R. Bailey, Lucy J. Goudswaard, Philip C. Haycock.

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
