## [Decision Letter · Decision Letter 0]

21 Jul 2025

Dear Dr. Hobson,

Thank you for submitting your manuscript to PLOS ONE. After careful consideration, we feel that it has merit but does not fully meet PLOS ONE’s publication criteria as it currently stands. Therefore, we invite you to submit a revised version of the manuscript that addresses the points raised during the review process.

We look forward to receiving your revised manuscript.

Kind regards,

Yazhou He

Academic Editor

PLOS ONE

Journal Requirements:

“LMH is supported in part by grant MR/N0137941/1 for the GW4 BIOMED MRC DTP, awarded to the Universities of Bath, Bristol, Cardiff and Exeter from the Medical Research Council (MRC)/UKRI

RMM is a National Institute for Health Research Senior Investigator (NIHR202411). RMM, LJG and PCH are supported by a Cancer Research UK 25 (C18281/A29019) programme grant (the Integrative Cancer Epidemiology Programme). RMM is also supported by the NIHR Bristol Biomedical Research Centre which is funded by the NIHR (BRC-1215-20011) and is a partnership between University Hospitals Bristol and Weston NHS Foundation Trust and the University of Bristol. Department of Health and Social Care disclaimer: The views expressed are those of the author(s) and not necessarily those of the NHS, the NIHR or the Department of Health and Social Care.

SERB was supported by an NIHR Advanced Fellowship (NIHR 301666) whilst undertaking this work. Additional support was provided by the Higgins family.”

4.Thank you for stating the following in the Competing Interests section:

“RMM, LJG and PCH have received funding from Cancer Research UK. LMH receives funding from the GW4 BioMed2 MRC DTP.”

We note that you received funding from a commercial source: Cancer Research UK and GW4 BioMed2 MRC DTP.

Additional Editor Comments:

This study presents a protocol of a potentially important contribution to cancer prognosis research.  

Reviewers' comments:

Reviewer's Responses to Questions

**Comments to the Author**

1. Does the manuscript provide a valid rationale for the proposed study, with clearly identified and justified research questions?

Reviewer #1: Yes

Reviewer #2: Yes

Reviewer #3: Yes

Reviewer #4: Yes

Reviewer #5: Yes

Reviewer #6: Yes

2. Is the protocol technically sound and planned in a manner that will lead to a meaningful outcome and allow testing the stated hypotheses?

Reviewer #1: Yes

Reviewer #2: Yes

Reviewer #3: Yes

Reviewer #4: Yes

Reviewer #5: Yes

Reviewer #6: Yes

3. Is the methodology feasible and described in sufficient detail to allow the work to be replicable?

Reviewer #1: Yes

Reviewer #2: Yes

Reviewer #3: Yes

Reviewer #4: Yes

Reviewer #5: Yes

Reviewer #6: No

4. Have the authors described where all data underlying the findings will be made available when the study is complete?

Reviewer #1: Yes

Reviewer #2: Yes

Reviewer #3: Yes

Reviewer #4: Yes

Reviewer #5: Yes

Reviewer #6: Yes

5. Is the manuscript presented in an intelligible fashion and written in standard English?

Reviewer #1: Yes

Reviewer #2: Yes

Reviewer #3: Yes

Reviewer #4: Yes

Reviewer #5: Yes

Reviewer #6: Yes

You may also provide optional suggestions and comments to authors that they might find helpful in planning their study.

Reviewer #1: Very interesting and overall fascinating work.

While the protocol provides a solid framework for Mendelian Randomization (MR) analysis, I recommend expanding the discussion in regards to horizontal pleiotropy as a potential methodological limitation. Pleiotropy, particularly when genetic instruments influence the outcome via pathways other than the exposure of interest, constitutes a major threat to the validity of causal inference in MR.

To strengthen the translational relevance and external validity of the colorectal cancer (CRC)-specific findings, I recommend that the authors consider incorporating a standalone Europe population based analysis before performing the analysis with the Asia Colorectal Cancer Consortium (ACCC).

Reviewer #2: This is a well-structured and timely protocol that proposes a novel application of polygenic risk scores (PRS) to identify circulating protein biomarkers associated with cancer development. The study is based on a solid theoretical framework, incorporates advanced analytical strategies, and is grounded on high-quality datasets such as UK Biobank and GWAS consortia.

The authors clearly articulate the rationale for using PRS as a proxy for genetic liability and justify the methodological choices with appropriate references. The distinction between forward and reverse causality is well addressed through the planned use of bidirectional Mendelian randomization, colocalization, and clustering approaches.

The methodology is technically robust, with transparent criteria for inclusion/exclusion of participants, detailed statistical procedures, and clear descriptions of software and tools to be used. The strategy to handle potential biases, including population stratification, horizontal pleiotropy, and survivorship bias, is particularly commendable.

The data management and sharing plan aligns with best practices in open science, and the authors demonstrate commitment to reproducibility through open code and future data availability. The language of the manuscript is precise, clear, and appropriate for a scientific audience.

In summary, this is a strong and comprehensive study protocol with the potential to generate valuable insights into cancer-related proteomics. I recommend its acceptance without revisions.

Reviewer #3: Greetings to the editor and the author. To be honest, the study protocol was very well planned. It took into account the scientific methods used and the rules of the PLOS ONE journal. When it comes to research ethics, the author must give the journal copies of the ethical approvals they got, in line with the timeframe of the research.

I would like to thank the editors and authors for their hard work and for the way they organised the scientific work.

Reviewer #4: I uploaded my review as an attachment.

I have no concerns or additional comments for the author on the research or publication ethics

Reviewer #5: This manuscript presents a study protocol to identify circulating protein biomarkers for early detection of colorectal and lung cancer using UK Biobank genetic data. Polygenic risk scores (PRS), derived through PRS-CS and PRS-CSx methods, are used to assess genetic liability to cancer and its association with 2,023 plasma proteins measured via the Olink platform. Bidirectional Mendelian Randomization (MR) distinguishes between causal, reverse, and non-causal relationships. Supporting analyses, including colocalization and time-to-diagnosis assessments, aim to refine causal inference and characterize protein changes relative to cancer onset. The study ultimately aims to improve early detection and risk prediction.

Strengths

-The study tackles an important and timely issue.

-The use of PRS-CS and PRS-CSx is a thoughtful and advanced methodological choice.

-The analysis plan is thorough, using multiple approaches like bidirectional Mendelian randomization and colocalization to strengthen causal inference.

-The study’s aims are clearly explained, with a solid rationale and potential for clinical benefit.

To further strengthen the manuscript, consider the following minor revisions;

Weaknesses

-The manuscript does not mention collider bias, which is a potential issue given the selection of prevalent cancer cases and volunteers from UK Biobank.

-The protocol does not assess whether subgroup analyses (e.g., by cancer subtype or smoking status) will be sufficiently powered to yield reliable results.

-There is no mention of using internal validation within UK Biobank or applying regularisation techniques to reduce overfitting, which is important given the high-dimensional proteomic data

-Writing Clarity:

The sentence “advantages of measuring protein within the blood are the reduced volume…” is grammatically awkward. Rephrasing it to “an advantage of measuring proteins in blood is the small volume required” would improve clarity.

• The sentence “five-year survival rates fall considerably…” is hard to follow. It should be rewritten with clearer structure and full sentence form for precision.

• Noun–verb mismatches, “advantages of measuring protein within the blood … are…”).

• The explanation sentence of Mendelian Randomisation and reverse Mendelian Randomisation can be more clarified.

Reviewer #6: Comments to Authors:

The authors present a study protocol that will use human genetic datasets to identify circulating proteins that may be causal, and thus used as early screening biomarkers, in the development of colorectal and lung cancer. There are many strengths to the proposed protocol, including cost and ease to implement in a clinical setting. Outcomes will include the identification or protein biomarkers associated with PRS for a given cancer, and protein changes may be more meaningful compared to gene changes when discussing functional biological processes. My specific comments are below:

Major Comments:

- The authors include an aim for the analyses, but not a clear aim for the study as a whole. The “overall aim of these analyses is to identify protein changes that are the causal consequence of genetic liability to cancer”. The hypothesis is that “protein level changes resulting from cancer development can be identified via an individual’s PRS for the disease, representing their genetic liability to developing that cancer”. The aim does not seem to match the hypothesis. Inferring the purpose of the study from the hypothesis suggests that the aim is to develop a protocol to determine if the genetic liability to cancer, identified using an individual’s PRS, is associated with measurable changes in circulating (blood plasma or serum) proteins. Then, the identified protein changes can be used to identify specific proteins that may be causal (forward or reverse) in the development of cancer. Please clarify.

- The text should be specific when referring to ‘disease’ and ‘cancer’. While the analyses may be relevant for different diseases and cancer types, the aims and hypothesis should be specific to the study – colorectal cancer and lung cancer.

- A workflow figure would be helpful in clarifying the protocol discussed in the methods section of the text. For example, summary statistics will be used from GWAS studies, then weights will be generated via PRS-CS, the O-link will be performed (on biobanked samples or will subjects be recruited and samples collected – not clear) …and so on.

- Include legends that describe what each figure is showing, annotate symbols, lines, colors, abbreviations etc.

Minor Comments:

- Indicate which panel is being discussed when describing the figures in the text. For example, figure 1 has three panels (A, B, C). When mentioning fig 1 in the text, it is not clear when authors are referring to a specific panel.

- Differentiate between PRS-CS and PRS-CSx. First line under methods and analysis section suggests that PRS-CS and PRS-CSx are synonymous acronyms.

- The text states that “forward and reverse Mendelian randomization sensitivity analyses, as well as colocalization analyses, will be performed….”. In the Strengths of the Study section, the authors state that a strength of the study is that the proposed method will increase the power compared to conventional MR. As currently written, it appears that the proposed method is ‘better’ than MR, but the proposed method still uses MR? Please clarify if MR will just be used to statistically show that the proposed method has more power compared to conventional MR.

- The text describes the assumptions shown figure 2, please make sure to include the description of IV1 (not annotated in figure), IV2, and IV3 either in the figure legend or a short description in the figure itself.

- A figure depicting what is discussed in the ‘study design’ section would be extremely helpful in clarifying the main points of the analyses. For example, depicting the predicted outcome for an association in the forward direction (protein is causal for the disease) or if association is found in the reverse direction (may suggest genetic liability to cancer is causing protein level change). Having simplified figures for these complex analyses will aid in communicating the science to broader populations.

- Please include additional details for performing data harmonization. For example, is this performed to ensure the same alleles are associated with the same effects across datasets and that all are coded in the same manner?

**Do you want your identity to be public for this peer review?** For information about this choice, including consent withdrawal, please see our Privacy Policy

Reviewer #1: No

Reviewer #2: **Yes: ** Érika Carvalho de Aquino

Reviewer #3: **Yes: ** Amran Ibrahim

Reviewer #4: No

Reviewer #5: No

Reviewer #6: No

---

## [Author Response · Author response to Decision Letter 1]

3 Oct 2025

We thank the reviewers for their constructive comments and feedback on the protocol. We have provided the reviewers comments in bold with responses to those comments bellow referencing the numbered lines on the track changes version of the revised manuscript.

Reviewer #1

Very interesting and overall fascinating work.

While the protocol provides a solid framework for Mendelian Randomization (MR) analysis, I recommend expanding the discussion in regards to horizontal pleiotropy as a potential methodological limitation. Pleiotropy, particularly when genetic instruments influence the outcome via pathways other than the exposure of interest, constitutes a major threat to the validity of causal inference in MR.

We agree that investigating horizontal pleiotropy is an important consideration in MR studies, lines 360 – 362 have been updated to reflect exclusion of the cis-pQTLs for the protein from reverse MR:

Reverse MR will be performed using SNPs associated with the cancer PRS at a threshold of p < 5 x 10-8 excluding cis-pQTLs for the protein.

Additionally, as stated on line 385, we will use MR methods to help account for this as well as colocalization to assess if the association is due to a shared causal variant.

The MR-PRESSO, weighted mode and weighted median methods will be used to assess IVs for horizontal pleiotropy […]. Colocalization analyses will be used to assess if genetic associations with cancer and proteins are due to shared causal variants at the same locus through genomic confounding.

We will explore the impact of this more in the results paper to follow but acknowledge the inherent issue of horizonal pleiotropy in MR studies.

To strengthen the translational relevance and external validity of the colorectal cancer (CRC)-specific findings, I recommend that the authors consider incorporating a standalone Europe population based analysis before performing the analysis with the Asia Colorectal Cancer Consortium (ACCC).

As stated, on lines 199 - 201 a standalone European subset of the colorectal cancer data will be used to make a European PRS and that the multi-ancestry score is an additional PRS.

In an effort to reduce the Eurocentric bias and to increase power in addition to developing a European PRS, we will be utilising GWAS summary statistics for colorectal cancer from European and East Asian ancestries to develop polygenic risk scores.

Reviewer #2

This is a well-structured and timely protocol that proposes a novel application of polygenic risk scores (PRS) to identify circulating protein biomarkers associated with cancer development. The study is based on a solid theoretical framework, incorporates advanced analytical strategies, and is grounded on high-quality datasets such as UK Biobank and GWAS consortia.

The authors clearly articulate the rationale for using PRS as a proxy for genetic liability and justify the methodological choices with appropriate references. The distinction between forward and reverse causality is well addressed through the planned use of bidirectional Mendelian randomization, colocalization, and clustering approaches.

The methodology is technically robust, with transparent criteria for inclusion/exclusion of participants, detailed statistical procedures, and clear descriptions of software and tools to be used. The strategy to handle potential biases, including population stratification, horizontal pleiotropy, and survivorship bias, is particularly commendable.

The data management and sharing plan aligns with best practices in open science, and the authors demonstrate commitment to reproducibility through open code and future data availability. The language of the manuscript is precise, clear, and appropriate for a scientific audience.

In summary, this is a strong and comprehensive study protocol with the potential to generate valuable insights into cancer-related proteomics. I recommend its acceptance without revisions.

Thank you for your comments recognising our rationale, methodology and open science goals.

Reviewer #3

Greetings to the editor and the author. To be honest, the study protocol was very well planned. It took into account the scientific methods used and the rules of the PLOS ONE journal. When it comes to research ethics, the author must give the journal copies of the ethical approvals they got, in line with the timeframe of the research.

I would like to thank the editors and authors for their hard work and for the way they organised the scientific work.

It was not necessary to obtain ethical approval for this study; the UK Biobank ethical approval letter was uploaded with the original protocol submission and detailed in the ethics section. Email confirmation has been received from GECCO allowing the use of the colorectal cancer site-specific GWAS data.

Reviewer #4

I uploaded my review as an attachment.

I have no concerns or additional comments for the author on the research or publication ethics

Thank you.

Reviewer #5

This manuscript presents a study protocol to identify circulating protein biomarkers for early detection of colorectal and lung cancer using UK Biobank genetic data. Polygenic risk scores (PRS), derived through PRS-CS and PRS-CSx methods, are used to assess genetic liability to cancer and its association with 2,023 plasma proteins measured via the Olink platform. Bidirectional Mendelian Randomization (MR) distinguishes between causal, reverse, and non-causal relationships. Supporting analyses, including colocalization and time-to-diagnosis assessments, aim to refine causal inference and characterize protein changes relative to cancer onset. The study ultimately aims to improve early detection and risk prediction.

Strengths

-The study tackles an important and timely issue.

-The use of PRS-CS and PRS-CSx is a thoughtful and advanced methodological choice.

-The analysis plan is thorough, using multiple approaches like bidirectional Mendelian randomization and colocalization to strengthen causal inference.

-The study’s aims are clearly explained, with a solid rationale and potential for clinical benefit.

To further strengthen the manuscript, consider the following minor revisions;

Weaknesses

-The manuscript does not mention collider bias, which is a potential issue given the selection of prevalent cancer cases and volunteers from UK Biobank.

We will delve into this further in the results paper to follow as this applies to the further analyses section. We also acknowledge the bias of the selection of individuals within the UKB in the limitations section specifically in regard to survivorship.

Prevalent cancer cases will reflect a specific subset of the general population with cancer, individuals who have survived cancer and were able to volunteer for the study; potentially introducing survivorship bias.

-The protocol does not assess whether subgroup analyses (e.g., by cancer subtype or smoking status) will be sufficiently powered to yield reliable results.

We will perform post-hoc power calculations to determine if there is sufficient power to yield results as we are using this approach as a discovery method.

-There is no mention of using internal validation within UK Biobank or applying regularisation techniques to reduce overfitting, which is important given the high-dimensional proteomic data

We do not believe this applies to this study design as we are not splitting the data and using it for training, we have used an external dataset independent of UK Biobank to generate PRS weights (GWAS that exclude the UK Biobank).

-Writing Clarity:

The sentence “advantages of measuring protein within the blood are the reduced volume…” is grammatically awkward. Rephrasing it to “an advantage of measuring proteins in blood is the small volume required” would improve clarity.

Corrected, see lines 106 – 108.

An advantage of measuring protein within the blood is the reduced volume of sample required for analysis and affordability, as opposed to other minimally invasive methods, such as circulating-tumour DNA (ctDNA) sequencing.

• The sentence “five-year survival rates fall considerably…” is hard to follow. It should be rewritten with clearer structure and full sentence form for precision.

Corrected, see lines 82 - 86.

Cancer is a leading cause of death worldwide (2) with survival rates considerably lower when diagnosis is made at a later stage. For colorectal cancer and lung cancer the 5-year survival is reduced from more than 9 in 10 and 6 in 10, respectively, when diagnosed at stage 1, to 1 in 10 for colorectal cancer and less than 1 in 10 for lung cancer when diagnosed at stage 4.

• Noun–verb mismatches, “advantages of measuring protein within the blood … are…”).

Corrected, see above.

• The explanation sentence of Mendelian Randomisation and reverse Mendelian Randomisation can be more clarified.

Corrected, see lines 354 – 357.

MR will be performed in the forward and reverse direction: forward MR, where the protein is the exposure and cancer is the outcome, will be used to estimate the effect of selected proteins on the cancer of interest and reverse MR, where cancer is the exposure and protein levels are the outcomes, will be used to estimate the effect of cancer liability on circulating protein concentration.

Reviewer #6

Comments to Authors:

The authors present a study protocol that will use human genetic datasets to identify circulating proteins that may be causal, and thus used as early screening biomarkers, in the development of colorectal and lung cancer. There are many strengths to the proposed protocol, including cost and ease to implement in a clinical setting. Outcomes will include the identification or protein biomarkers associated with PRS for a given cancer, and protein changes may be more meaningful compared to gene changes when discussing functional biological processes. My specific comments are below:

Major Comments:

- The authors include an aim for the analyses, but not a clear aim for the study as a whole. The “overall aim of these analyses is to identify protein changes that are the causal consequence of genetic liability to cancer”. The hypothesis is that “protein level changes resulting from cancer development can be identified via an individual’s PRS for the disease, representing their genetic liability to developing that cancer”. The aim does not seem to match the hypothesis. Inferring the purpose of the study from the hypothesis suggests that the aim is to develop a protocol to determine if the genetic liability to cancer, identified using an individual’s PRS, is associated with measurable changes in circulating (blood plasma or serum) proteins. Then, the identified protein changes can be used to identify specific proteins that may be causal (forward or reverse) in the development of cancer. Please clarify.

Aims updated, see lines 160 – 164.

Cancer early detection remains a challenge, with a lack of specific biomarkers for lung and colorectal cancer. This research aims to identify protein level changes that are causally downstream of genetic liability to lung cancer and colorectal cancer for use as potential biomarkers for these cancers. To achieve this, we will use a combination of PRS, observational analyses, bidirectional Mendelian Randomization and colocalization approaches.

- The text should be specific when referring to ‘disease’ and ‘cancer’. While the analyses may be relevant for different diseases and cancer types, the aims and hypothesis should be specific to the study – colorectal cancer and lung cancer.

Updated throughout.

- A workflow figure would be helpful in clarifying the protocol discussed in the methods section of the text. For example, summary statistics will be used from GWAS studies, then weights will be generated via PRS-CS, the O-link will be performed (on biobanked samples or will subjects be recruited and samples collected – not clear) …and so on.

See figure 3.

- Include legends that describe what each figure is showing, annotate symbols, lines, colors, abbreviations etc.

Updated, see figures.

Minor Comments:

- Indicate which panel is being discussed when describing the figures in the text. For example, figure 1 has three panels (A, B, C). When mentioning fig 1 in the text, it is not clear when authors are referring to a specific panel.

Corrected throughout.

- Differentiate between PRS-CS and PRS-CSx. First line under methods and analysis section suggests that PRS-CS and PRS-CSx are synonymous acronyms.

Corrected, see lines 196 – 199.

PRS-CSx applies the same methodology as PRS-CS to multi-ancestry GWAS summary statistics, improving generalisability of results to more ancestry groups within the global majority by allowing the use of different ancestry GWAS summary statistics and LD panels for those populations.

- The text states that “forward and reverse Mendelian randomization sensitivity analyses, as well as colocalization analyses, will be performed….”. In the Strengths of the Study section, the authors state that a strength of the study is that the proposed method will increase the power compared to conventional MR. As currently written, it appears that the proposed method is ‘better’ than MR, but the proposed method still uses MR? Please clarify if MR will just be used to statistically show that the proposed method has more power compared to conventional MR.

Updated, see lines 46 - 53.

- Strengths of the study:

o A strength of using PRS in the discovery step of identifying proteins to carry through into MR, is that lifetime genetic liability to cancer is captured with the use of a continuous measure of genetic risk. In contrast, using cancer registry data only captures those who already have a diagnosis and not those who may go on to be diagnosed imminently.

o This study will use a novel approach to construct PRS using weights generated from SNPs across the genome, we expect this to be more powerful for discovery in comparison with a PRS constructed using only genome-wide significant SNPs.

- The text describes the assumptions shown figure 2, please make sure to include the description of IV1 (not annotated in figure), IV2, and IV3 either in the figure legend or a short description in the figure itself.

Updated, see figure 2.

- A figure depicting what is discussed in the ‘study design’ section would be extremely helpful in clarifying the main points of the analyses. For example, depicting the predicted outcome for an association in the forward direction (protein is causal for the disease) or if association is found in the reverse direction (may suggest genetic liability to cancer is causing protein level change). Having simplified figures for these complex analyses will aid in communicating the science to broader populations.

Figure 1 illustrates this, updated figure legend to provide further clarification.

- Please include additional details for performing data harmonization. For example, is this performed to ensure the same alleles are associated with the same effects across datasets and that all are coded in the same manner?

Updated, see lines 412 - 414.

Harmonisation across the GWAS summary statistics datasets will be performed using the TwoSampleMR R package to ensure that the effect on exposure and outcome are in the same direction, using effect allele frequency (EAF) to infer the strand, palindromic SNPs are kept where they are inferable from EAF.

---

## [Decision Letter · Decision Letter 1]

11 Nov 2025

A protocol for using human genetic data to identify circulating protein level changes that are the causal consequence of cancer processes.

PONE-D-24-41052R1

Dear Dr. Hobson,

We’re pleased to inform you that your manuscript has been judged scientifically suitable for publication and will be formally accepted for publication once it meets all outstanding technical requirements.

Kind regards,

Yazhou He

Academic Editor

PLOS ONE

Additional Editor Comments (optional):

Reviewers' comments:

Reviewer's Responses to Questions

**Comments to the Author**

1. Does the manuscript provide a valid rationale for the proposed study, with clearly identified and justified research questions?

Reviewer #5: Yes

Reviewer #6: Yes

2. Is the protocol technically sound and planned in a manner that will lead to a meaningful outcome and allow testing the stated hypotheses?

Reviewer #5: Yes

Reviewer #6: Yes

3. Is the methodology feasible and described in sufficient detail to allow the work to be replicable?

Reviewer #5: Yes

Reviewer #6: Yes

4. Have the authors described where all data underlying the findings will be made available when the study is complete?

Reviewer #5: Yes

Reviewer #6: Yes

5. Is the manuscript presented in an intelligible fashion and written in standard English?

Reviewer #5: Yes

Reviewer #6: Yes

You may also provide optional suggestions and comments to authors that they might find helpful in planning their study.

Reviewer #5: All major and minor reviewer comments have been adequately addressed. Fix the study design typo “willl be defined” to “will be defined.”

Reviewer #6: Thank you for addressing each of the comments. The addition of figure three strengthens the manuscript.

**Do you want your identity to be public for this peer review?** For information about this choice, including consent withdrawal, please see our Privacy Policy

Reviewer #5: No

Reviewer #6: No

---

## [Editor Report · Acceptance letter]

PONE-D-24-41052R1

PLOS ONE

Dear Dr. Hobson,

I'm pleased to inform you that your manuscript has been deemed suitable for publication in PLOS ONE. Congratulations! Your manuscript is now being handed over to our production team.

Kind regards,

on behalf of

Dr. Yazhou He

Academic Editor

PLOS ONE